# *Ry_chc_* Confers Extreme Resistance to *Potato virus Y* in Potato

**DOI:** 10.3390/cells11162577

**Published:** 2022-08-18

**Authors:** Gege Li, Jingjing Shao, Yuwen Wang, Tengfei Liu, Yuhao Tong, Shelley Jansky, Conghua Xie, Botao Song, Xingkui Cai

**Affiliations:** 1Key Laboratory of Potato Biology and Biotechnology, Ministry of Agriculture and Rural Affairs, Huazhong Agricultural University, Wuhan 430070, China; 2Key Laboratory of Horticultural Plant Biology, Ministry of Education, Huazhong Agricultural University, Wuhan 430070, China; 3Department of Horticulture, University of Wisconsin, Madison, WI 53706, USA

**Keywords:** potato, PVY, extreme resistance, *Ry_chc_*, evolutionary analysis

## Abstract

The *Potato virus Y* (PVY) is responsible for huge economic losses for the potato industry worldwide and is the fifth most consequential plant virus globally. The main strategies for virus control are to limit aphid vectors, produce virus-free seed potatoes, and breed virus-resistant varieties. The breeding of PVY-resistant varieties is the safest and most effective method in terms of cost and environmental protection. *Ry_chc_*, a gene that confers extreme resistance to PVY, is from *S. chacoense*, which is a wild diploid potato species that is widely used in many PVY-resistant breeding projects. In this study, *Ry_chc_* was fine mapped and successfully cloned from *S. chacoense* accession 40-3. We demonstrated that *Ry_chc_* encodes a TIR-NLR protein by stably transforming a diploid susceptible cultivar named AC142 and a tetraploid potato variety named E3. The *Ry_chc_* conferred extreme resistance to PVY^O^, PVY^N:O^ and PVY^NTN^ in both of the genotypes. To investigate the genetic events occurring during the evolution of the *Ry_chc_* locus, we sequenced 160 *Ry_chc_* homologs from 13 *S. chacoense* genotypes. Based on the pattern of sequence identities, 160 *Ry_chc_* homologs were divided into 11 families. In Family 11 including *Ry_chc_*, we found evidence for Type I evolutionary patterns with frequent sequence exchanges, obscured orthologous relationships and high non-synonymous to synonymous substitutions (Ka/Ks), which is consistent with rapid diversification and positive selection in response to rapid changes in the PVY genomes. Furthermore, a functional marker named MG64-17 was developed in this study that indicates the phenotype with 100% accuracy and, therefore, can be used for marker-assisted selection in breeding programs that use *S. chacoense* as a breeding resource.

## 1. Introduction

Potato (*Solanum tuberosum*), the world’s fourth most important food crop, after maize, wheat and rice, is highly adaptable and produces high yields. However, potato diseases, especially viruses, severely decrease the production of potatoes. *Potato virus Y* (PVY) is one of the most widespread viruses infecting potatoes and is responsible for severe economic losses. PVY is responsible for yield losses of up to 80%, and also negatively affects tuber quality [1]. Breeding cultivars resistant to PVY provides the most straightforward and economical solution to prevent the damage caused by PVY [2]. The *R* genes can provide durable and extreme resistance to PVY, and thus can be used for breeding resistance to PVY. In the same way as other RNA viruses, the genetic diversity of PVY is rich because of high rates of evolution and mutation, resulting in diverse and complex strains [3]. PVY consists of five non-recombinant strains (PVY^O^, PVY^C^, PVY^N^, PVY^Z^, PVY^E^) and many recombinant genotypes, such as PVY^N:O^ and PVY^NTN^ [4]. In recent years, recombinant-like strains, such as PVY^NTN^, were reported to cause tuber necrotic ringspot disease and were extensively studied worldwide [5].

The potato plants may exhibit compatible or incompatible interactions with the virus. Both the sensitive and tolerant potato plants are susceptible to PVY. The susceptibility leads to the virus replicating and spreading throughout the plant in a compatible interaction. After accumulating large amounts of virus, the sensitive potatoes show signs of disease. In contrast, the resistant potatoes exhibit no or only very mild symptoms. The three types of virus resistance that result from incompatible interactions are extreme resistance (ER), hypersensitive resistance (HR) and susceptibility genes (S-genes)-conferred resistance. The ER and HR are dominant resistance while the S-genes-conferred resistance is recessive [6].

The potatoes with dominant resistance to PVY are easy to uncover, maintain and are more popular in breeding programs. Although the 10 PVY-dominant resistance genes were localized to individual chromosomes, only a small percentage of the genes were cloned and studied at the DNA sequence level [2]. The hypersensitivity-type resistance genes include *Ny-1* [7] and *Ny-Smira* [8] on Chromosome IX, *Ny-2* [9] on Chromosome XI and *Ny_tbr_* [10] and *Ny_sp_*_l_ [11] on Chromosome IV. The five extreme resistance genes that were mapped are *Ry_adg_* [12] and *Ry_sto_* (XI) [13] on Chromosome XI, *Ry_sto_* [14] and *Ry-f_sto_* [15] genes on the short arm of chromosome XII and *Ry_chc_* [16] on Chromosome IX. So far, only one gene, *Ry_sto_*, with extreme resistance to PVY, was cloned from the wild relative, *S. stoloniferum*. They found that the EDS1-SAG101-NRG1 node is required to execute the *Ry_sto_*-initiated immunity. In plants, *Ry_sto_* interacts directly with PVY CP and the correct folding of the CP pore is essential for interactions. The *Ry_sto_* can recognize the CPs of a minimum of 10 important viruses with a similar core region and enable *Solanaceae* and *Brassicaceae* resistance against the plum pox virus and turnip mosaic virus [2,17]. The *Ry_chc_* on Chromosome IX from the wild species *S. chacoense* has not been cloned.

A study demonstrated that most of the NBS-LRR genes are organized as tight and complex clusters in the genome. The clustering may contribute to the rapid evolution of *R* genes in the plant genome by facilitating the recombination of homologous genes [18]. Previous work on *R*-genes in lettuce, rice and *Solanaceae* species indicates that the particular *R*-genes can be divided into two types, Type I and Type II resistance genes [19,20,21]. Type I resistance genes are characterized by high frequency sequence exchanges, obscured orthologous relationships and high non-synonymous to synonymous substitutions (Ka/Ks) between or within a species. In contrast, the type II resistance genes are characterized by a rather slow rate of evolution and rare sequence exchanges. However, the copy number variation and the evolutionary patterns present within the *Ry_chc_* locus remain unclear. 

The phenotypic evaluation of the breeding process for PVY resistance is a time intensive process, because this process depends on phenotypic characterization experiments that usually require PVY virus inoculation, or take place in the field under high disease pressure conditions. The molecular markers that can indicate the phenotype of plants provide a good solution to this problem, because they are stable, can be used regardless of the season and environment and do not require viral inoculation and identification.

In this study, we report on the map-based cloning of the *Ry_chc_* gene, using a diploid population that was segregating for resistance to PVY. We prepared this population by crossing the PVY-resistant *S. chacoense* accession 40-3 with the PVY-susceptible *S. berthaultii* accession 143-6. We used the Infinium 8303 Potato SNP array to identify the SNPs linked to the *Ry_chc_* gene in the F1 population. Utilizing high-density genetic mapping, we localized the gene to a small region at the distal end of chromosome 9. We demonstrated that *Ry_chc_* encodes a TIR-NLR protein using a map-based cloning strategy and that *Ry_chc_* confers extreme resistance to PVY in the diploid potato cultivar AC142, and the tetraploid potato variety E3. The identification of *Ry_chc_* provides unique genetic resources for the study of PVY resistance and PVY-resistance breeding. To further understand the origin and evolutionary history of this locus, we sequenced many DNA fragments encoding the NBS and LRR domain of *Ry_chc_* from 13 *S. chacoense* genotypes. These data indicate a Type I evolutionary pattern for *Ry_chc_*, which is characterized by large variations in copy number, frequent sequence exchanges and high Ka/Ks values, especially in the region encoding the LRR domain. We also produced a gene marker named MG64-17 that is useful for marker-assisted selection to accelerate the breeding process of PVY resistance in the breeding programs that use *S. chacoense* as a source of PVY resistance.

## 2. Materials and Methods

### 2.1. Plant Material and Phenotypic Characterization

A diploid population that is known as a pseudo-testcross and that consisted of 3711 F1 plants was developed by crossing a wild diploid *S. chacoense* accession 40-3 (accession PI 320285) female with a wild diploid *S. berthaultii* accession 143-6 (accession PI 558129) male. The two parental lines used in the study differed in their responses to PVY. The *S. chacoense* accession 40-3 is resistant to PVY. In contrast, the *S. berthaultii* accession 143-6 is susceptible to PVY. A total of 175 plants were tested for their response to PVY and were used for the gene mapping. Additionally, 3536 plants were screened for close recombinants that were used for the fine mapping of the *Ry_chc_* gene. The phenotype of the close recombinants was also tested. Each genotype that was used for phenotypic identification was represented by eight plants.

The PVY^O^ strain was activated in tobacco that grew to the six-leaf stage and was found to not have other viruses. After the activation of the PVY^O^ strain was completed, it was inoculated mechanically, using an airbrush set (Paasche model H airbrush; Chicago, IL, USA). Five leaves from the different shoots on the top of the plants were sampled to test for the presence of PVY, using RT-PCR at two weeks after inoculation. Four weeks later, the same method was used to collect tissue for the ELISAs. The plants that tested negatively with the ELISAs were re-inoculated. Four weeks later, the ELISAs were performed again. The PVY^O^ was detected, using the double-antibody sandwich method (DAS-ELISA), as described previously [22].

### 2.2. SNP Array Development and SNP Genotyping

A total of 142 plants with stable phenotypes were selected for the DNA extraction, using a modified cetyltrimethylammonium bromide (CTAB) method and stored at −20 °C prior to array genotyping. The DNA concentration was adjusted to 50 ng/µL. We genotyped 142 F1 plants on an Illumina iScan Reader, utilizing the Infinium 8303 Potato Array (Illumina Inc., San Diego, CA). We analyzed the results using the Illumina GenomeStudio software (Illumina, San Diego, CA) [23]. 

### 2.3. Map Construction and Linkage Mapping 

Prior to mapping, the SNPs with incorrect or missing genotypes and low-quality SNPs in the parental lines and F1 were removed. The progeny with more than 17 (12%) missing genotypes were also deleted. The remaining SNP markers were used for linkage analysis. The genetic map was constructed using JoinMap4.0 (Kyazma, Wageningen, The Netherlands) [24]. The population types considered for the segregation analysis were type 1 lm × ll, locus heterozygous in the first parent *S. chacoense* accession 40-3 (P1), two or three alleles; type 2 nn × np, locus heterozygous in the second parent *S. berthaultii* accession 143-6 (P2), two or three alleles; type 3 hk × hk, locus heterozygous in two parents *S. chacoense* accession 40-3 (P1) and *S. berthaultii* accession 143-6 (P2), three or four alleles. We chose regression mapping with Haldane’s function as a calculation option, and constructed the linkage groups with a LOD score from 3 to 10. The visualization of the paternal and maternal genetic linkage mapping was achieved using the MapChart V2.3 (Wageningen University and Research, Wageningen, The Netherlands) [25]. The markers linked to *Ry_chc_* were identified from the statistically significant differences in the distribution of the number of resistant plants from the two different genotypes, which was calculated with a *t*-test.

### 2.4. Development of Molecular Markers

Based on the DM1-3 reference genome sequence (PGSC v4.03), pseudomolecules and BAC sequence of the *S. chacoense* accession 40-3 [26,27], we designed the PCR-based DNA markers using the Primer Premier 5.0 software (PREMIER Biosoft International, San Francisco, CA, USA). A total of 20 PVY-resistant progeny and 20 PVY-sensitive progeny from a population of 175 individual plants were selected for constructing a resistant pool (R-pool) and a susceptible pool (S-pool). The PCR-based markers that indicated the polymorphisms between the R-pool and the S-pool were used to genotype the F1 progeny from the *S. chacoense* accession 40-3 × *S. berthaultii* accession 143-6 cross. All of the tightly linked markers are listed in Appendix A. The PCRs were performed in 20 μL volumes. The reaction mixtures included 1 μL of genomic DNA (50 ng/μL), 10 μL of 2 × UTaq PCR MasterMix (Beijing Zoman biotechnology Co,. Ltd., Beijing, China), 1 μL each of the forward and reverse primers (10 mM), and 7 μL of the double distilled water. The PCR conditions were as follows: denaturation at 94 °C for 3 min; 36 cycles of denaturation at 94 °C for 30 s, annealing at 54 °C for 30 s, extension at 72 °C for 30 s; a final extension at 72 °C for 10 min, and cooling to 12 °C. All of the PCR products were analyzed using 40% polyacrylamide gel electrophoresis.

### 2.5. BAC Library Construction, Screening and Sequencing

A Bacterial Artificial Chromosome library (EIGHTSTAR BIO-TECH, Wuhan, China) was constructed, using the PVY-resistant parent *S. chacoense* accession 40-3. The 74,880 BAC clones were stored in a total of 195 384-well plates. The average insert size was approximately 130 kb. All of the clones on the 384-well plates were mixed into a super pool, yielding 195 super pools. The target super pools were screened from the 195 target mixed pools, using the left flank of the closely linked markers (M2526, M28, M71-20, M71-21-1 and M71-28-3) and the right flank markers (SNP48, M4950 and M50). All of the clones in each row and each column of each target super pool were mixed into a row and column pool. The corresponding markers were used to locate the target clones. The positive BAC clone plasmids for sequencing were purified, using the QIAGEN Large-Construct Kit (Qiagen, Hamburg, Germany). The sequencing was entirely performed using the PacBio RS II sequencing platform, and then assembled into one contig. Then, the 71-24H and 74-24B were sequenced by Personalbio Biotechnology Co., Ltd. (Shanghai, China), 100-8O by EIGHTSTAR BIO-TECH (Wuhan, China) and 64-14E by Frasergen Information Co. (Wuhan, China).

### 2.6. Recombinants and High-Density Genetic Map for Ry_chc_

The initial flanking random DNA markers, M4 and M32, were selected to screen the recombinants among the 1078 F1 individuals. Then, M28 and M50 were subsequently used to screen the recombinants among the 2458 F1 individuals. The phenotypic identification of the recombinants was the same as described for the population. Chi-squared (χ2) tests for goodness of fit were performed to determine the agreement of the observed frequencies of the genotypes with the theoretical Mendelian separation ratios. JoinMap4.0 (Kyazma, Wageningen, The Netherlands) was used with default parameters to construct a high-density genetic map. The linkage to *Ry_chc_* was determined using the Kosambi mapping and a LOD score of 3.0 as a threshold [24]. The genetic linkage map was drawn with the software Mapchart V2.3 (Wageningen University and Research, Wageningen, The Netherlands) [25]. The start and end positions of each marker were averaged for assigning their physical location in the genome. 

### 2.7. Potato Transformation

The sequences of the candidate *Ry_chc_* genes *C3* and *C4* were amplified from BAC64-14E with the Phanta Super-Fidelity DNA Polymerase (Vazyme, Nanjing, China), following the manufacturer’s recommendations. To generate OAC-C3 and OAC-C4 transgenic plants, 3940-bp and 4167-bp DNA fragments from the start codon to the stop codon were cloned into pBI121, which contains a 35S promoter. The pBI121 was digested with XbaI and SacI, using the ClonExpress® II One Step Cloning Kit (Vazyme, Nanjing, China), according to the manufacturer’s instructions. Three vectors were introduced into the Agrobacterium tumefaciens strain GV3101, using the heat shock method, which were then used to transform the AC142. A 8940-bp genomic DNA fragment containing all of the exons and introns of *C4*, a 3752-bp promoter sequence, a 1015-bp downstream region and a Sal I site between the promoter and start codon was cloned into pBI121 and digested with HindIII and SacI, using the ClonExpress® II One Step Cloning Kit (Vazyme, Nanjing, China), according to the manufacturer’s instructions. This vector was introduced into the *Agrobacterium tumefaciens* strain GV3101 and used to transform the tetraploid potato variety E3, using an Agrobacterium-mediated transformation method [28]. All of the primers used for amplification during the Agrobacterium-mediated transformation procedure are listed in Appendix A.

### 2.8. RNA Extraction and RT-qPCR

The total RNA was extracted using a Plant Total RNA Kit (ZOMANBIO, ZP405-1, Beijing, China), according to the manufacturers’ instructions. The first-strand cDNA was synthesized using a five × All-In-One RT MasterMix (with AccuRT Genomic DNA Removal Kit) Reverse Transcription Kit (Applied Biological Materials Inc., Richmond, Canada). The gene and PVY expression was quantified by RT-qPCR using a LightCycler 480 II (Roche Diagnostics, IN, USA) and EvaGreen two × qPCR Master Mix (Applied Biological Materials, Vancouver, Canada). The potato Ef1a gene was used as an internal control for the data normalization [29]. The expression levels of the genes were calculated using the 2^−ΔCq^ method described by Bio-Rad (Hercules, CA, USA). All of the primers used for RT-qPCR analysis are listed in Appendix A.

### 2.9. Phylogenetic Analysis

The specific primers were designed to amplify the paralogs of *Ry_chc_* from two resistant wild accessions of *S. chacoense* and two susceptible wild accessions of *S. chacoense*. Based on the four *Ry_chc_* homologs from the four wild accessions of *S. chacoense*, the conserved primers were designed to amplify the *Ry_chc_* gene fragments from the genomic DNA of *S. chacoense* accession 40-3, the resistant parent and 12 additional *S. chacoense* genotypes. Three independent reactions were performed for each genotype. The groups of six additional clones were sequenced using the conserved primer, until there were no singletons. One clone from each group of identical sequences in each genotype was then sequenced. The clones of the fragments were named to reflect their genotypes. The last digit indicates the number of the clone. The letter or number before the last digit is an abbreviation of the species name. The conserved NBS and LRR motifs were predicted using the Pfam database (http://pfam.xfam.org/search#tabview=tab0, accessed on 17 July 2022).

The putative genomic fragments from each wild accession of *S. chacoense* were aligned, using MAFFT with default parameters [30]. The nucleotide identity between the two sequences was calculated using BioEdit. The phylogenetic trees were generated with MEGA X, first using the Neighbor-joining statistical method with 1,000 bootstrap replicates [31]. We then used FastTree with the Jukes–Cantor model to construct the ML phylogenetic tree [32]. All of the trees were exported as improvable graphs online by iTOL [33].

The nucleotide coding sequences (CDSs) of all of the genomic fragments were predicted using the Fgenesh database from Softberry (http://www.softberry.com/berry.phtml?topic=fgenesh&group=programs&subgroup=gfind, accessed on 17 July 2022) [34]. To detect the positive selection, the ratios of nonsynonymous to synonymous nucleotide substitutions (Ka/Ks) were calculated with MEGA X using the full-length coding sequences (CDSs) and the core regions of the LRR domains predicted by Pfam, which is regarded as the determinant of recognition specificity for the Avr factors. Seven methods with default parameters from the RDP4 software, including 3Seq, Bootscan, Chimaera, GENECONV, MaxChi, RDP and SiScan, were used to investigate the sequence exchange between the *Ry_chc_* genes [35,36,37,38,39,40,41,42].

## 3. Results

### 3.1. Construction of a High-Density Intraspecific SNP Map for Potato

To clone an *R* gene, named *Ry_chc_*, we constructed a segregating population of diploids by crossing the PVY-resistant *S. chacoense* accession 40-3 with the PVY-susceptible *S. berthaultii* accession 143-6. In the F1 generation, 91 individuals were resistant to PVY and 84 were susceptible. These data are consistent with the resistance phenotype segregating in a 1:1 ratio (*χ*^2^ = 0.206 *p* = 0.65), and with a single dominant gene conferring resistance to PVY in this population, which is consistent with previous studies [16]. We used the Infinium 8303 Potato SNP array to find the SNPs linked to the *Ry_chc_* gene in the F1 population, which contained 142 individuals. The array-based genotypic results showed that a total of 8282 SNPs were scored with normal signals among the 142 *S. chacoense* accession 40-3 × *S. berthaultii* accession 143-6 F1 individuals. After removing the SNPs with incorrect or missing genotypes in the parental lines and the F1 samples, the remaining 304 SNPs were used to construct a genetic map. Consequently, a genetic map that consists of 237 SNP markers in 160 recombination bins was constructed, with an average bin interval of 5.74 cM. This CB-SNP map was 918.82 cM and consisted of 12 linkage groups (Figure 1; Appendix A).

Among the remaining 237 SNP markers included in the CB-SNP map, six SNPs that are located on the distal end of chromosome 9 were identified as significantly correlated with the *Ry_chc_* gene (Appendix A). Based on these SNPs and the recombinant A128, we mapped *Ry_chc_* to a 2.4-Mb region on chromosome 9 (PGSC v4.03) (Figure 2a).

### 3.2. Map-Based Cloning of Ry_chc_

Using Primer Premier 5.0, we developed 431 random PCR-based markers in the genes that were predicted using the software from SoftBerry (http://www.softberry.com/berry.phtml?topic=fgenesh&group=programs&subgroup=gfind, accessed on 17 July 2022) and that are located in the 2.4-Mb interval. Then, 20 resistant individuals and 20 susceptible individuals were mixed into the R-pool and S-pool, respectively, to perform BSA using these markers. A total of 10 markers, namely M4, M2425-2, M2526, M28, M4950-1, M4950-2, M50, M32, M152 and M236, were potentially linked to *Ry_chc_*. When there was no polymorphism in the amplification products of the resistant materials and susceptible materials, their PCR products were directly sequenced to search for specific SNPs. Using this approach, we successfully obtained one SNP marker named SNP48 (Figure 2b). A total of 142 individuals were used to verify the linkage between the markers and genes. The results proved that these markers are truly linked to *Ry_chc_*. Based on the recombinants A99 and A128, *Ry_chc_* was localized to a chromosome region of approximately 113,352 bp between M2526 and SNP48 (PGSC v4.03) (Figure 2b).

To further narrow the *Ry_chc_* region, the flanking markers M28 and M50 were used for genotyping a fine-mapping population composed of 2,458 F1 offspring from the *S. chacoense* accession 40-3 × *S. berthaultii* accession 143-6 cross. Five close recombinants were identified, namely Re1, Re2, Re3, Re4 and Re5. We inoculated all of the five plants with PVY^O^-FL and found that, although Re1 and Re2 were susceptible to PVY, the other plants were resistant (Appendix A). Using these five new recombinants and recombinant A99, we narrowed the *Ry_chc_* gene to a 107,664-bp interval (PGSC v4.03) between M28 and SNP48 (Figure 2b). However, no linkage markers were developed between M28 and SNP48, probably due to the many sequence differences between the reference genome and the disease-resistant parent, or possibly because of the poor quality of the reference genome sequence in this region, which contains a total of six gaps in this region.

To solve the problem of the reference genome, a BAC library with approximately 10-fold coverage of the potato genome was constructed, using the resistant parent *S. chacoense* accession 40-3. The left-flanking markers M2526 and M28 and the right-flanking markers SNP48, M4950 and M50 were used to screen the BAC library, and eight positive BAC clones were obtained, which we named 71-24H, 74-24B, 100-8O, 114-11B, 133-21L, 164-22I, 170-20H and 170-20I. Three of the BAC clones (71-24H, 74-24B and 100-8O) that provide the most coverage of the fine mapping region were selected for sequencing. A total of 32 pairs of random DNA markers were designed, using the newly sequenced BAC sequences, and three pairs of markers named M71-20, M71-21-1 and M71-28-3, were linked to *Ry_chc_*. However, since this group of BAC clones did not completely cover the entire region, we used marker M71-28-3 to identify more of the BAC clones. Three BAC clones, named 64-14E, 96 and 147, were identified. The clone 64-14E was sequenced. The sequence demonstrated that all three of the sequenced BAC clones failed to completely cover the fine mapping interval. The primers were designed using the disease-susceptible BAC clone 100-8O. The missing part of the interval containing the locus was amplified and sequenced. The final assembly result showed that the physical distance between M28 and SNP48 was 174,054 bp (Figure 2c).

After obtaining the sequence of the fine mapping interval from *S. chacoense* accession 40-3, we subsequently developed six pairs of polymorphic markers that were tightly linked to the resistant phenotype and, therefore, allowed us to map *Ry_chc_* to a 27,273-bp interval between M71-28-3 and M64-17 using the recombinants P86 and A99 (Figure 2c). The sequence analysis of this interval, using FGENESH on the Softberry website, predicted a total of six genes in the region. The conserved structural domains of the proteins encoded by the genes in this interval were predicted using the Conserved Domain database on the NCBI website. Based on these data, the gene we named *C2* (*Candidate*
*2*) encodes an oxidoreductase, the genes we named *C3* and *C4* encode TIR-NLR class proteins, and the genes we named *C1*, *C5* and *C6* encode proteins with no conserved domains (Figure 2d; Appendix A).

### 3.3. C4 Is Ry_chc_ and Is Responsible for PVY Resistance

Most of the *R* genes identified so far are NBS-LRR-type resistance genes that encode the proteins containing a nucleotide binding domain (NBD) and a leucine-rich repeat (LRR) domain. Therefore, we focused on *C3* and *C4*, which encode the TIR-NBS-LRR proteins. The full-length genomic sequences were amplified from BAC64E14 and using homologous recombination, were cloned into pBI121, which contains a CaMV 35S promoter. The susceptible diploid potato cultivar AC142 was separately transformed with each candidate gene using the Agrobacterium-mediated genetic transformation. Three stable transgenic plants were obtained for each candidate gene, respectively (Appendix A). The transgenic plants were inoculated with PVY^O^-FL. Approximately one month after inoculation, the non-inoculated leaves of AC142 and the transgenic plants harboring the *C3* gene became severely wrinkled and mottled. In contrast, we observed no symptoms of PVY^O^-FL infection in the non-inoculated leaves of the transgenic plants harboring the *C4* gene (Figure 3). Two weeks later, the virus was detected in the plants using a RT-qPCR-based assay (Figure 4a). Four weeks later, the virus was detected using ELISA (Figure 4c). We found that following inoculation, the virus did not replicate or spread in the transgenic lines containing the *C4* candidate gene. In contrast, we found that large amounts of virus accumulated in the transgenic plants containing the *C3* gene. Indeed, the AC142 and transgenic plants that contained the *C3* gene accumulated similar amounts of virus (Appendix A). 

To further study the resistance phenotype, the PVY-sensitive tetraploid potato variety E3 was transformed with *Ry_chc_*, including 3,752 bp of upstream sequence and 1,015 bp of the downstream sequence. After inoculating with PVY^O^, the data from the RT-qPCR assay (Figure 4b) and ELISA (Figure 4d) indicated that the virus did not accumulate in the systemic leaves of the transgenic lines. Approximately one month after inoculation with PVY^O^, the leaves of the E3 plants developed a mosaic pattern and appeared crimpled. In contrast, none of the transgenic plants developed any symptoms (Appendix A). These results indicate that *C4* confers a resistance to PVY^O^-FL in both the AC142 and E3. Moreover, the data indicate that the *C4* candidate gene is the PVY resistance gene, named *Ry_chc_*. 

### 3.4. Ry_chc_ Expression Restricts Systemic Spread of a Range of PVY Strains 

To test whether *Ry_chc_* was resistant to different PVY strains, the transgenic plants harboring *Ry_chc_* and derived from AC142 and E3, including OAC-C4-1, OAC-C4-2, OAC-C4-3 and OE3-C4-1, OE3-C4-2 and OE3-C4-3 were inoculated with PVY^N:O^ and PVY^NTN^. Two weeks after inoculation with the virus, we tested whether the virus was present in the top non-inoculated leaves of the plants, using qPCR. We found that the transgenic plants were similar to the *S. chacoense* accession 40-3 (i.e., the resistant parent) in that neither of the two viruses accumulated. In contrast, both of the viruses accumulated significantly in AC142 and E3 (Figure 5a,b). The data from the analysis of the top non-inoculated leaves at four weeks after inoculation using ELISAs were the same as the results of the qPCR assays (Figure 5c,d). The virus did not spread in either the transgenic plants or in the *S. chacoense* accession 40-3. In contrast, the two different viruses caused systemic infection in AC142 and E3. These data demonstrate that the plants harboring *Ry_chc_* were resistant to both the PVY^N:O^ and PVY^NTN^. The results demonstrate that *Ry_chc_* is not dependent on specific genotypes of potato to provide resistance to PVY.

### 3.5. Evolution of the Ry_chc_ Locus 

To study the evolutionary pattern of *Ry_chc_* in additional wild accessions of *S. chacoense*, we first conducted a phenotypic identification of the 12 accessions of *S. chacoense*. We found that 6 of the 12 accessions were resistant to PVY and that the remaining six accessions were susceptible (Figure 6c). The *Ry_chc_* gene encodes TIR-NBS-LRR proteins with a total number of 13 LRRs (Figure 6a). We used primers specific to the *Ry_chc_* gene to amplify paralogs of *Ry_chc_* from the two wild accessions of *S. chacoense* that are resistant to PVY and the two wild accessions of *S. chacoense* that are susceptible to PVY. The PCR products were cloned, and at least 12 replicates of each clone were sequenced. Based on the sequences of the four *Ry_chc_* homologs from the four wild accessions of *S. chacoense*, the conserved primers were designed to amplify *Ry_chc_* gene fragments containing the NBS and LRR regions from the genomic DNA of the *S. chacoense* accession 40-3, the resistant parent and the 12 additional *S. chacoense* genotypes. A total of 160 distinct gene fragments were amplified from the 13 genotypes. The copy number varied from four to 20 in each genotype (Appendix A). These data indicate that a rapid copy number evolution may have occurred in these genes. We noticed that distinct gene fragments were present in the different accessions. Finally, we found 133 unique gene fragments in total and that the number of LRRs varied from 0 to 17, which is consistent with the interpretation that the *Ry_chc_* genes are members of a large cluster of homologous genes. 

We used MAFFT with default parameters to perform an alignment of 133 different sequences from 13 genotypes and found that large indels (>156 bp) were mainly concentrated in the intron and exon regions that encode the LRRs. We analyzed the 25 fragments that have large insertion sequences in the exon region and found that the deletions were also nonrandom. We found deletions in part of exon 3 and the entirety of exon 4 in only four fragments. The deletions in the remaining 21 fragments were mainly concentrated within the middle 284 bp of exon 4. The length of the missing sequences ranges from 156 bp to 1147 bp and encode LRRs. These deletions remove from 3 to 11 LRR domains (Table 1). We found that the hot spot for unequal crossing over was in the middle of exon 5 and that this hot spot was responsible for generating these large deletions. Further analysis of all 26 fragments revealed that the point mutations causing amino acid changes were also the leading cause of the variations in LRR number.

We used ML (Appendix A) and NJ (Appendix A) methods to build the phylogenetic trees for the 160 fragments. Seven homologs of *Ry_chc_* from Solanaceaeas, including *N-like_Caann* from the pepper genome, *TAO1_Solyc* and *N-like_Sopen* from the tomato genome, and *N-like_Niatt*, *N-like_Nisyl*, *TAO1-like_Nitab* and *TAO1-like_Nitom* from the tobacco genome were used as the outgroup. The topological structure of the ML phylogenetic tree was similar to that of the NJ phylogenetic tree. We used the ML phylogenetic tree for further analysis, because of its higher bootstrap value. According to the patterns of sequence diversity among the genes, we divided the tree into 11 distinct clades. The bootstrap values that separate the 11 clades are 100%, 100%, 98.8%, 98.7%, 100%, 100%, 94.9%, 100%, 99.6%, 100% and 99.6%. We performed a further analysis of Family 11, which includes the fragment of *Ry_chc_*. The sequence identities of exons 2, 3 and 4 were 92.7%, 94.6% and 80.5%, respectively. In contrast, the sequence similarities of introns 2, 3 and 4 were only 90.7%, 90.4% and 83.3% (Figure 6d). The sequence identity of the NBS region ranged from 83.3% to 100% (mean = 93.8%). Among the 1,653 pairwise nucleotide identities, 1,140 were not lower than 95%. In contrast, the sequence identities of the LRR region ranged from 7.6% to 100% (mean = 81.9%). Moreover, only 426 of the 1,653 pairwise nucleotide identities were not lower than 95% (Figure 6d). Thus, the sequence similarities of the LRR regions were lower than the sequence similarities of the NBS regions with a wider range of variation.

In Family 11, there was no clear relationship between the taxonomy and position to determine the possible homologous relationships between genes. Most of the Ka/Ks ratios for the CDS of Family 11 were all less than 1, which indicates a negative selection pressure. However, the majority of the Ka/Ks ratios in the LRR region were greater than 1, which provides reliable evidence of positive selection (Figure 6e). We further investigated possible recombination events in this family to study its role in evolution. The recombination events of Family 11 were tested simultaneously, using seven methods—3Seq, Bootscan, Chimaera, GENECONV, MaxChi, RDP and SiScan. The results showed that the number of recombination events may range from 20, predicted by Chimaera, to 34, predicted by GENECONV (Appendix A). We analyzed the tight clade containing *Ry_chc_* and found that it contained nine genes, all from the PVY-resistant accessions. Only the *S. chacoense* accession 40-3 and CHC39-7 had two genes in this tight clade. The remaining five accessions had only one gene (Figure 6b). The sequence alignments revealed that 40-3-12 should have evolved through a gene recombination between *Ry_chc_* and a gene from another family. All of the pairwise sequence identities among the remaining eight genes in this tight clade were more than 99%. The high levels of sequence identity may be responsible for the PVY resistance of these five accessions (Appendix A).

### 3.6. Marker Development for Breeding

The breeding of cultivars resistant to PVY provides the simplest and most economical solution for preventing crop damage from PVY [2]. *S. chacoense* is a good breeding resource, because of its male fertility and because either the female or male parents can be used as a source of *Ry*. Indeed, *Ry_chc_* from *S. chacoense* has been used in Japanese and Russian potato breeding programs [43,44].

To test whether the markers developed in this study can be used for molecular marker-assisted selection for breeding, three markers M71-20, MG64-17 and M64-17 were used to investigate the genotype of 35 accessions kept in our laboratory. We found that MG64-17 is predictive of *Ry_chc_*, in that a specific band can be amplified only in resistant accessions (Figure 7; Appendix A). Therefore, MG64-17 can be used as a marker for the marker-assisted selection of PVY resistance in breeding programs when *S. chacoense* is used as a source of PVY resistance. The detection technique is simple and can accurately identify the PVY-resistant plants without phenotypic characterization, which requires either inoculation with PVY or the growth of plants in high disease-pressure conditions.

## 4. Discussion

In the preliminary mapping experiments with a CB-SNP map, we localized *Ry_chc_* to the distal end of chromosome 9, which is consistent with previous studies [16]. These data indicate that the chromosomal location of *Ry_chc_* differs from the chromosomal locations of other extreme resistance genes—*Ry_adg_*, *Ry_sto_* and *Ry-f_sto_*. A large cluster of 19 candidate genes between 60 Mb and 61 Mb are homologous to the tomato tospovirus-resistant gene Sw-5 [45]. In addition, the structure of the genomic fragments of *S. chacoense* on which *Ry_chc_* was located on may be quite different relative to the reference genome, DM1-3. These circumstances create challenges for the development of the gene-linked markers. The construction of a BAC library from the resistant parent is an ideal solution to this problem [46]. In this work, *S. chacoense* accession 40-3 served as the resistant parent and was used to construct a BAC library. We screened a BAC library, sequenced four BAC clones and assembled a 297.7-kb contig that covered the entire interval that we defined with fine mapping experiments. Based on the assembled contig, we found that the interval between M28 and SNP48 is 174,054 bp in the *S. chacoense* accession 40-3 and is 107,664 bp in DM1-3. These data demonstrate that the interval defined by our fine-mapping experiments differs significantly between the disease-resistant parent and the reference genome, as was previously speculated. After determining that the reference genome and the genome of the *S. chacoense* accession 40-3 were different at the *Ry_chc_* locus, we found a novel PVY extreme resistance gene *Ry_chc_* and that the genomic sequence identities between *Ry_chc_* and *Ry_sto_* are only 41.51%. Additionally, *Ry_sto_* is located on chromosome XII (Appendix A) [2]. 

The breeding resistance to the many different strains of PVY is a major goal of breeding engineering. We found that *Ry_chc_* can confer extreme resistance to PVY^O^, PVY^N:O^ and PVY^NTN^ in both the susceptible diploid cultivar AC142 and in the tetraploid potato variety E3. Moreover, *Ry_chc_* can confer resistance to PVY in potatoes from different genotypes, and therefore, these data indicate that *Ry_chc_* can confer resistance to PVY regardless of the genotype. Based on these data, we conclude that *Ry_chc_* is an ideal gene for breeding resistance to PVY in potato.

The genes that encode NBS-LRR proteins are numerous, of ancient origin and are most of the disease resistance genes in plants. Indeed, the genes that encode NBS-LRR proteins are one of the largest disease-resistant gene families known in plants [47]. We found at least 133 homologous fragments of *Ry_chc_* that contain NBS and LRR regions in the 13 *S. chacoense* accessions, which indicate that the *Ry_chc_* genes are members of a large cluster of homologous genes. Our analysis revealed that the sequence identity of intron 4 is much lower than intron 2 and intron 3. Similar to the situation with the introns, the sequence identity of exon 4 was much lower than the other two exons, which may be the result of diversity selection. In contrast to *Ry_chc_*, diversity selection is not the main evolutionary force of the *RGC2* genes in lettuce and frequent sequence exchanges homogenized the introns of the gene family [21].

Our analysis of 133 fragments indicates that a variety of genetic events contributed to the diversity of *Ry_chc_* genes. The recombinations, insertions, deletions and point mutations appeared to change the gene copy number and the number of LRR repeats. These data demonstrate a Type I evolutionary pattern for the *Ry_chc_* gene similar to the *RGC2* genes in lettuce, *RPP8* genes in Arabidopsis, *Pi2/9* homologs in rice and *RB*/*Rpi*-*blb1* genes in Solanaceae [19,20,21,48]. Two different types of evolution were observed in *RGC2*, *RPP8*, *Pi2/9* and *RB*/*Rpi*-*blb1*. Type I is a type of fast sequence evolution that is characterized by numerous sequence exchanges. Type II is a type of slow evolution with few sequence exchanges. Interestingly, the Type I evolutionary pattern that we observed with the *Ry_chc_* genes is characterized by numerous sequence exchanges, high Ka/Ks ratios and nucleotide divergence of CDS, especially in the LRR region. The main function of the LRR proteins is to participate in specific protein–protein interactions and in either a direct or indirect way, to recognize pathogens [49]. The complex evolution of the *Ry_chc_* genes leads to a large number of *Ry_chc_* homologs, and, therefore, may serve as an effective strategy to help potatoes cope with PVY, which is a diverse and continually evolving threat. 

The allelic variation created by the mutation and genetic recombination between alleles and family members are the main drivers of disease resistance gene evolution [50]. The late blight-resistant gene *Rpi-blb1* probably evolved from the sequence exchange between the ancestral genes *RGA3-blb* and *RGA1-blb* [51]. In our study, we found that the sequence exchange between *Ry_chc_* and a gene from another family led to the evolution of the *Ry_chc_* gene homolog 40-3-12. Moreover, 39-7-28/133, the potential functional ortholog of *Ry_chc_*, accumulated nonsynonymous point mutations in the course of long-term evolution. All of the evidence proves that *Ry_chc_* is a relatively old gene.

## 5. Conclusions

In this study, we cloned a novel PVY extreme resistance gene *Ry_chc_* that encodes a TIR-NLR protein and evolved with Type I evolutionary patterns.

## Figures and Tables

**Figure 1 cells-11-02577-f001:**
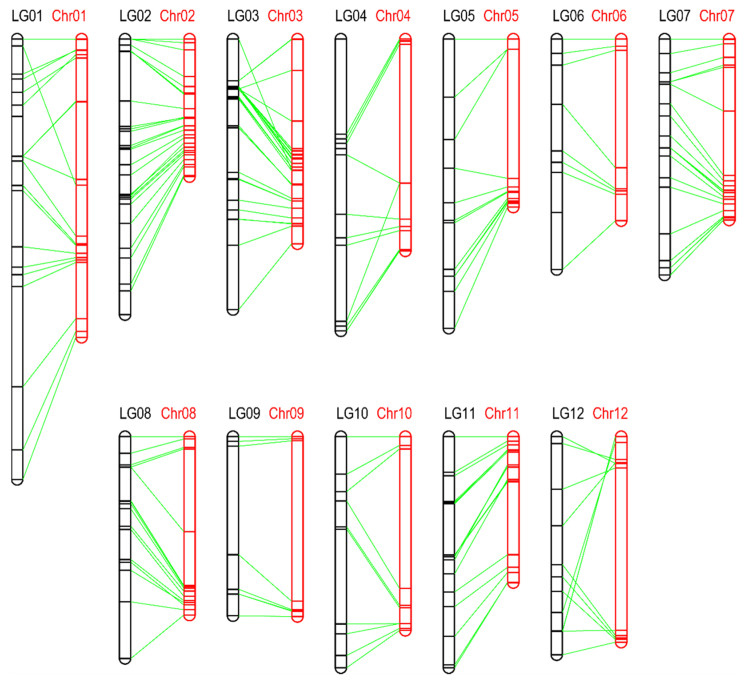
Potato CB-SNP genetic map and comparison with the DM1-3 genome physical map. LG1–LG12 (indicated with black) indicate 12 linkage groups constructed with 258 SNP markers. Chr1–Chr12 (indicated with red) indicate 12 potato chromosomes on the physical map. The synteny between the genetic and physical maps is showed with green lines.

**Figure 2 cells-11-02577-f002:**
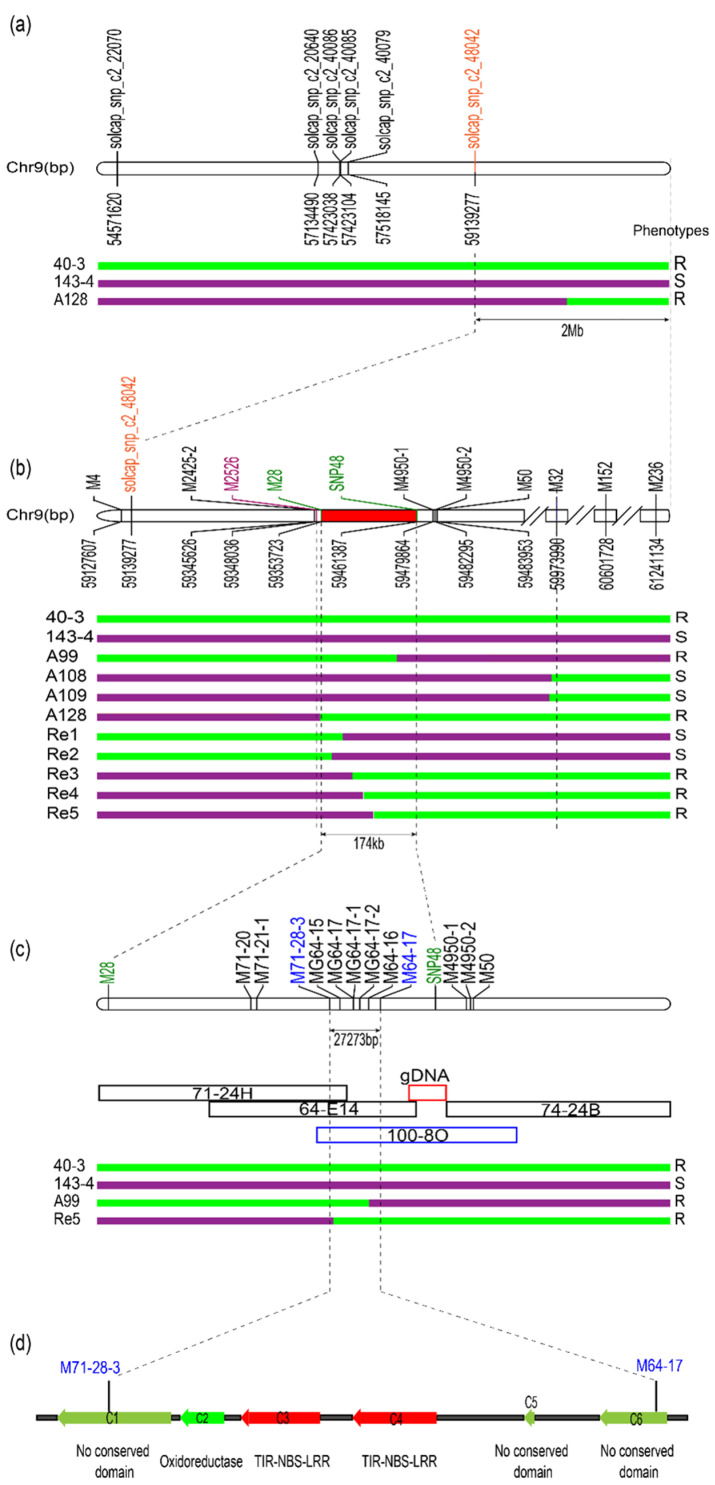
Map-based cloning of *Ry_chc_* in potato. (**a**) Preliminary mapping of *Ry_chc_*. *Ry_chc_* was preliminarily mapped to the end of chromosome 9 using six SNPs that were significantly linked to *Ry_chc_* and recombinant A99; (**b**) Fine mapping of *Ry_chc_*. The *Ry_chc_* locus was fine mapped to a 113,351-bp interval between M28 and SNP48. Markers indicated with a bold font were used for BAC library screening; (**c**) Physical locations of gene markers. The physical map was designed based on the sequence of a BAC clone from a library constructed from *S. chacoense* accession 40-3 and used for the fine-mapping of the *Ry_chc_* locus. The positions of the BAC clones in the fine mapping interval are indicated; (**d**) Schematic representation of 6 candidate genes between marker M71-28-3 and M64-17 in the genome of *S. chacoense* accession 40-3. The genes were predicted using FGENESH from the Softberry website. The C in genes’ name indicates candidate. Their conserved domains were predicted using the Conserved Domain database at NCBI. The transition between green and purple rectangles in (**a**–**c**) indicates that the cross-over happened in the progenies.

**Figure 3 cells-11-02577-f003:**
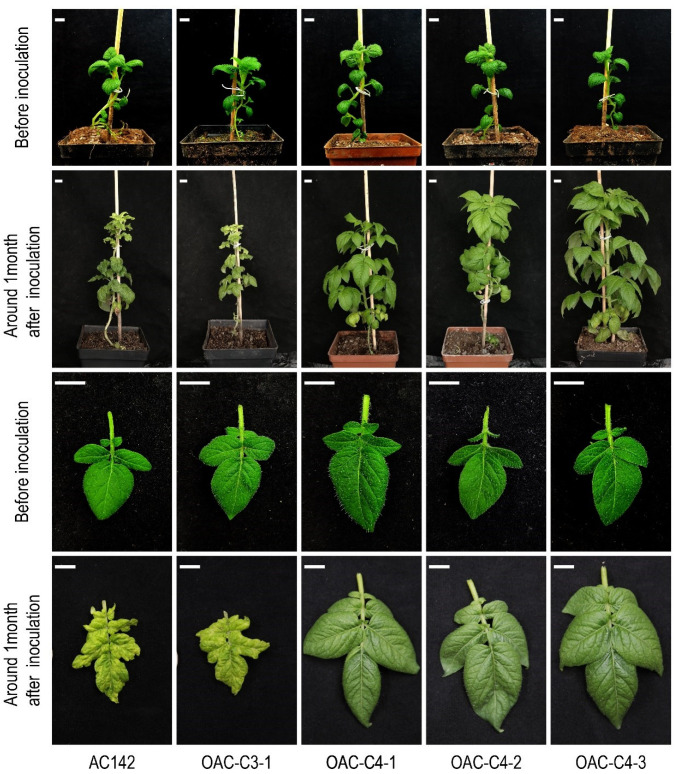
Symptoms of transgenic plants derived from AC142 at approximately 1 month after inoculation with PVY^O^. Representative plants and leaves are shown.

**Figure 4 cells-11-02577-f004:**
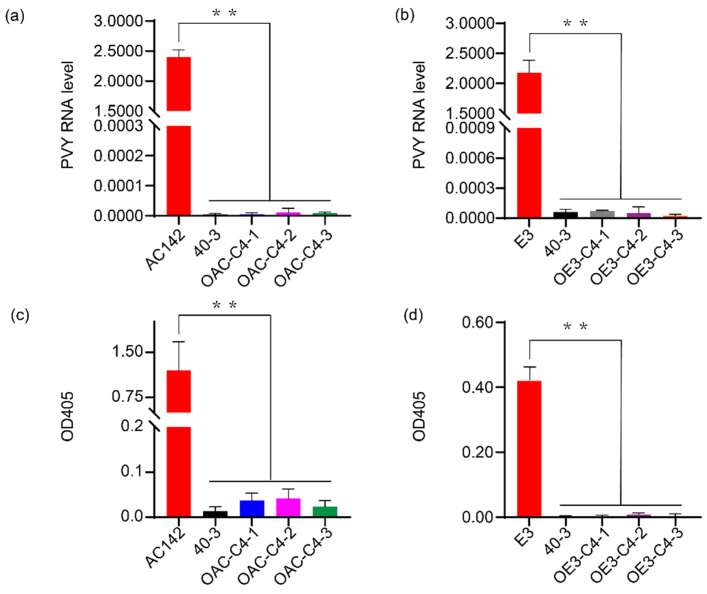
Phenotyping of PVY^O^ resistance in transgenic lines derived from AC142 and E3. (**a**) PVY RNA levels in the non-inoculated leaves of AC142 transgenic plants harboring *C4* two weeks after PVY^O^-FL inoculation. 40-3 indicates *S. chacoense* accession 40-3, the resistant parent; (**b**) ELISA titers of PVY^O^-FL in transgenic plants derived from AC142 harboring *C4* four weeks after inoculation. An ELISA (ODA405) value <0.1 indicates that the plants were resistant. An ELISA (ODA405) value >0.1 indicates that the plants were susceptible. Data in (**a**,**c**) are presented as mean values ± SE. ** indicates a statistically significant difference (*p* < 0.01) compared to AC142, calculated using a Student’s *t*-test. Data in (**b**,**d**) are presented as mean values ± SE. ** indicates a statistically significant difference (*p* < 0.01) compared to E3 calculated using a Student’s *t*-test.

**Figure 5 cells-11-02577-f005:**
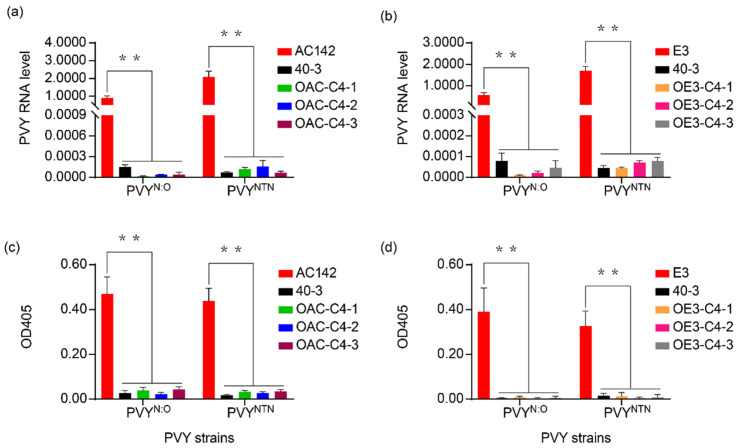
Phenotyping of extreme resistance in transgenic lines derive from AC142 and E3. (**a**) PVY RNA levels in the non-inoculated leaves of transgenic lines derived from AC142 and harboring *C4.* RNA levels were quantified two weeks after inoculation with PVY^N:O^ and PVY^NTN^. The 40-3 indicates *S. chacoense* accession 40-3, the resistant parent; (**b**) PVY RNA levels in the non-inoculated leaves of transgenic lines. RNA levels were quantified two weeks after inoculation with PVY^O^-FL, PVY^N:O^ and PVY^NTN^; (**c**) Titers of PVY^N:O^ and PVY^NTN^ in transgenic lines derived from AC142 and harboring *C4.* Titers were determined using ELISAs four weeks after inoculation; (**d**) Titers of PVY^O^, PVY^N:O^ and PVY^NTN^ in transgenic lines derived from E3 and harboring *C4.* Titers were determined using ELISAs four weeks after inoculation. The 40-3 indicates *S. chacoense* accession 40-3, the resistant parent. Data in (**a**) and (**c**) are presented as mean values ± SE. ** indicates a statistically significant difference (*p* < 0.01) compared to AC142 as determined using a Student’s *t*-test. Data in (**b**) and (**d**) are presented as mean values ± SE. ** indicates a statistically significant difference (*p* < 0.01) compared to E3 as determined using a Student’s *t*-test.

**Figure 6 cells-11-02577-f006:**
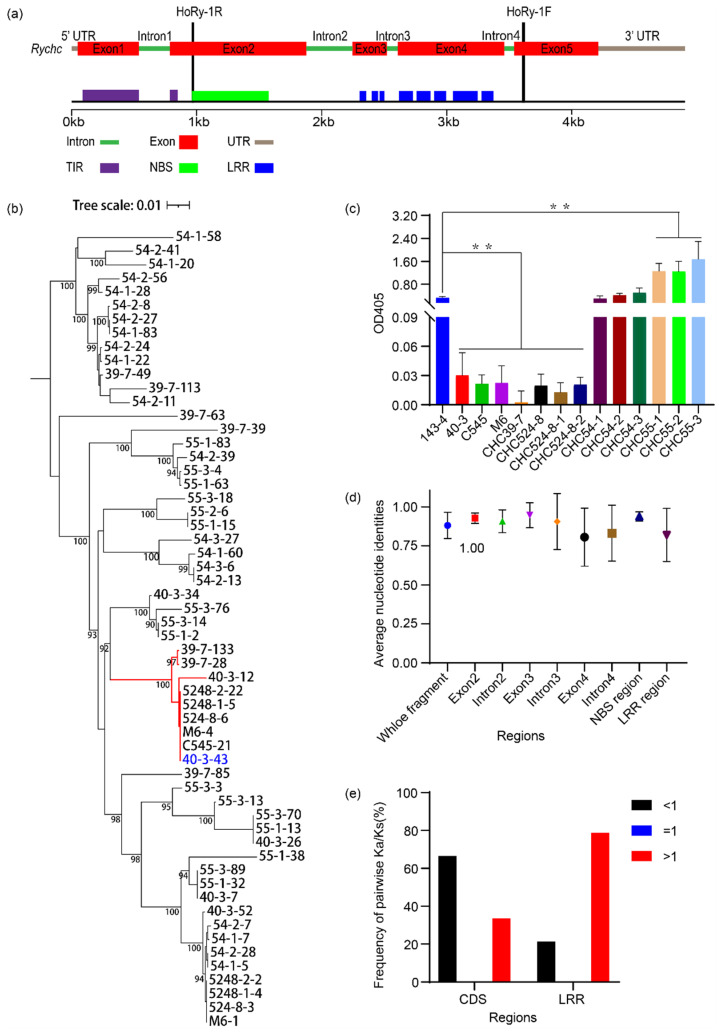
Evolution of *Ry_chc_* homologs in diverse wild accessions of *S. chacoense*. (**a**) Genomic structure and conserved domains of *Ry_chc_*; (**b**) Maximum Likelihood (ML) phylogenetic tree of the *Ry_chc_* homologs from Family 11. 40-3-43 in blue is the fragment of *Ry_chc_*; (**c**) Phenotyping of PVY resistance in 12 accessions of *S. chacoense*. Data in (**c**) are presented as mean values ± SE. ** indicates a statistically significant difference (*p* < 0.01) relative to 143-6, as determined using a Student’s *t*-test. The 40-3 indicates *S. chacoense* accession 40-3, the resistant parent. The 143-6 indicates *S. berthaultii* accession 143-6, the susceptible parent; (**d**) Average nucleotide identities of different parts of all *Ry_chc_* genes fragments; (**e**) Distribution of Ka/Ks values in the CDS and LRR regions of *Ry_chc_* genes from Family 11.

**Figure 7 cells-11-02577-f007:**
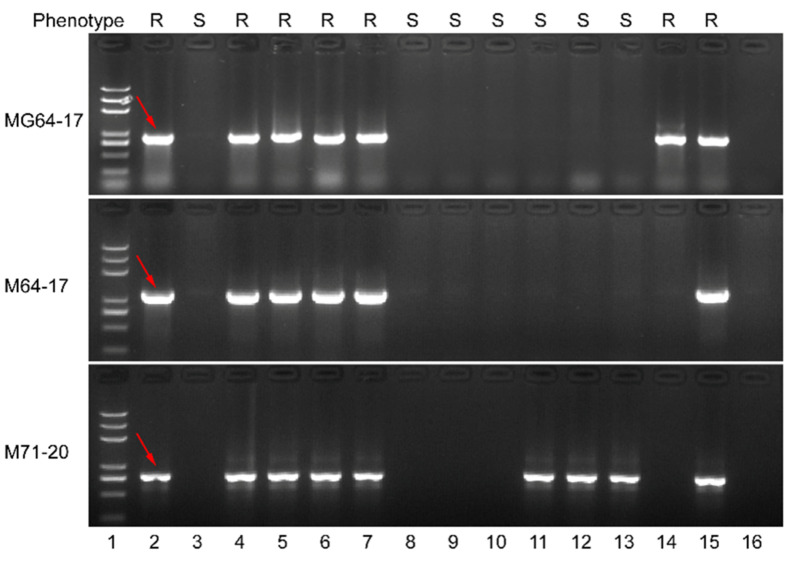
Detection of MG64-17, M64-17and MG71-20 in 12 *S. chacoense* lines. Lanes 1-16: Trans2K®Plus DNA Marker; *S. chacoense* accession 40-3; *S. berthaultii* accession 143-6; M6; CHC524-8; CHC524-8-1; CHC524-8-2; CHC54-1; CHC54-2; CHC54-3; CHC55-1; CHC55-2; CHC55-3; CHC39-7; C545; no template control. One of the specific bands that can be amplified only in resistant accessions are indicated with red arrows.

**Table 1 cells-11-02577-t001:** Large deletions and point mutations in the LRR regions.

Fragment Names	No. of All LRRs	Ka of LRR	No. of Deleted LRRs	Deletion Positions				Deletion Size
				Start	End	Start	End	
54-2-46	0	-	12	1444	2590			1147
54-3-5	0	0.077	12	1444	2470			1027
54-3-27	0	-	12	1444	2590			1147
54-3-2	0	0.36	12	1447	1517	1553	2602	1121
39-7-3	7	0.148	3	1955	2139			185
39-7-14	7	0.148	3	1955	2139			185
39-7-119	7	0.148	3	1955	2139			185
39-7-125	7	0.148	3	1955	2139			185
54-2-6	7	0.148	3	1955	2139			185
40-3-7	8	0.101	3	2083	2208	2381	2410	156
55-1-32	8	0.101	3	2083	2208	2381	2410	156
40-3-52	8	0.123	3	2083	2208	2381	2410	156
C545-4	6	0.107	3	2083	2208	2381	2410	156
M6-1	6	0.107	3	2083	2208	2381	2410	156
524-8-3	6	0.107	3	2083	2208	2381	2410	156
5248-1-4	6	0.107	3	2083	2208	2381	2410	156
5248-2-2	6	0.107	3	2083	2208	2381	2410	156
54-1-5	8	0.123	3	2083	2208	2381	2410	156
54-1-7	8	0.123	3	2083	2208	2381	2410	156
54-2-7	8	0.123	3	2083	2208	2381	2410	156
54-2-28	8	0.123	3	2083	2208	2381	2410	156
54-2-45	9	0.116	3	2083	2208	2381	2410	156
55-1-38	9	0.106	3	2083	2208	2381	2410	156
55-3-29	10	0.144	3	2083	2208	2381	2410	156
55-3-89	8	0.101	3	2083	2208	2381	2410	156

Positions of the deletions are compared to fragment 40-3-43(part of *Ry_chc_*).

## Data Availability

Sequence data have been deposited in the GenBank under the following accession numbers: the sequence of 100-8O, 74-24B, 71-24H and the missing part that was not covered by BAC, ON553906 to ON553909; homologous gene fragments from *S. chacoense* accession40-3, ON553745 to ON553759; homologous gene fragments from CHC39-7, ON553760 to ON553779; homologous gene fragments from C545, ON553780 to ON553787; homologous gene fragments from M6, ON553788 to ON553792; homologous gene fragments from CHC524-8, ON553793 to ON553798; homologous gene fragments from CHC524-8-1, ON553799 to ON553802; homologous gene fragments from CHC524-8-2, ON553803 to ON553808; homologous gene fragments from CHC54-1, ON553809 to ON553828; homologous gene fragments from CHC54-2, ON553829 to ON553846; homologous gene fragments from CHC54-3, ON553847 to ON553854; homologous gene fragments from CHC55-1, ON553855 to ON553874; homologous gene fragments from CHC55-2, ON553875 to ON553884; homologous gene fragments from CHC55-3, ON553885 to ON553904.

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
