# Peer review of "Rychc Confers Extreme Resistance to Potato virus Y in Potato"

_cells, 2022, doi:10.3390/cells11162577_

Round 1
Reviewer 1 Report
In this manuscript, the authors describe the map-based cloning of the Ry-chc gene for extreme resistance to Potato Virus Y. Very few genes have been cloned that confer resistance to this economically important virus family (Ry-sto, Pvr2) and this adds to the novelty and impact of this work. The experimental design is sound and the approaches are well thought-out and appropriate. The activity of this gene was verified using transgenic plants in multiple backgrounds and with multiple virus strains. Overall, the manuscript is well written and easy to understand, with only a few minor grammatical errors. I have noted a few below.
L39: change 'that' to 'of'
L89: segregating for resistance to PVY
L155: listed in
L549: is not dependent on
L604: occurred
L190, L202: since C3, C4, and E3 have not been defined yet (methods are listed before results), it would be helpful to say something like "The sequences of candidate Rychc genes C3 and C4 were amplified..." and "...used to transform tetraploid potato cultivar E3 using..."
Author Response
Thank you for your decision and constructive comments on my manuscript. We have carefully considered your suggestions on grammatical errors. We have carefully revised the whole text, including the 6 points pointed by you in the manuscript. We tried our best to correct the grammatical error. Please see the attachment. We hope you are satisfied with our modifications.

Reviewer 2 Report
The authors of the MS investigated a gene (Rychc) found in a wild diploid potato species (S. chacoense) and frequently used for the development of PVY resistance in cultivated potatoes. Rychc was fine-mapped and successfully cloned from S. chacoense accession 40-3 in this study. By stably transforming a diploid susceptible cultivar named AC142 and a tetraploid potato variety named E3, the author demonstrated that Rychc encodes a TIR-NLR protein. In both genotypes, Rychc conferred extreme resistance to PVYO, PVYN:O, and PVYNTN. The authors also looked into the genetic events that occurred during the evolution of the Rychc locus and sequenced 160 Rychc homologs from 13 different S. chacoense genotypes. 160 Rychc homologs were classified into 11 families based on the pattern of sequence identities. They discovered Type I evolutionary patterns in Family 11, including Rychc, with frequent sequence exchanges, obscured orthologous relationships, and high non-synonymous to synonymous substitutions (Ka/Ks), which is consistent with rapid diversification and positive selection in response to rapid changes in the PVY genomes. Furthermore, in this study, a functional marker called MG64-17 was developed that accurately indicates the phenotype and can thus be used for marker-assisted selection in breeding programmes that use S. chacoense as a breeding resource. Before commencing the trials, a thorough background was presented in the MS using a high level of English. All experiments were meticulously planned and executed with precision. The techniques used in the study were very effectively applied, and the results are very pleasing. The findings are extremely well supported and backed up by relevant references. Overall, the manuscript was well-written and presented new data that should be of interest to those researching potato viruses, making it suitable for publication. Every table and figure is useful and of high quality. The gel photos were created with great skill.
Author Response
Thank you very much for your comprehensive summary and praise.